# Timing matters in the use of renin-angiotensin system modulators and COVID-related cognitive and cerebrovascular dysfunction

**Mackenzi Meier[1], Sara Becker[1], Erica Levine[1], Oriana DuFresne[1], Kaleigh Foster[2], Joshua Moore[2], Faith N. Burnett[2], Veronica C. Hermanns[2], Stan P. Heath[2], Mohammed Abdelsaid[2], Maha Coucha[3]***

1 Department of Pharmacy Practice, School of Pharmacy, South University, Savannah, Georgia, United States of America, 2 Department of Biomedical Sciences, School of Medicine, Mercer University, Savannah, Georgia, United States of America, 3 Department of Pharmaceutical Sciences, School of Pharmacy, South University, Savannah, Georgia, United States of America

* mcoucha@southuniversity.edu

**Data Availability Statement:** All relevant data are within the paper and its Supporting Information files.

## Abstract

Renin-angiotensin system (RAS) modulators, including Angiotensin receptor blockers (ARB) and angiotensin-converting enzyme inhibitors (ACEI), are effective medications for controlling blood pressure. Cognitive deficits, including lack of concentration, memory loss, and confusion, were reported after COVID-19 infection. ARBs or ACEI increase the expression of angiotensin-converting enzyme-2 (ACE-2), a functional receptor that allows binding of SARS-CoV-2 spike protein for cellular invasion. To date, the association between the use of RAS modulators and the severity of COVID-19 cognitive dysfunction is still controversial. Purpose: This study addressed the following questions: 1) Does prior treatment with RAS modulator worsen COVID-19-induced cerebrovascular and cognitive dysfunction? 2) Can post-treatment with RAS modulator improve cognitive performance and cerebrovascular function following COVID-19? We hypothesize that pre-treatment exacerbates COVID-19-induced detrimental effects while post-treatment displays protective effects. Methods: Clinical study: Patients diagnosed with COVID-19 between May 2020 and December 2022 were identified through the electronic medical record system. Inclusion criteria comprised a documented medical history of hypertension treated with at least one antihypertensive medication. Subsequently, patients were categorized into two groups: those who had been prescribed ACEIs or ARBs before admission and those who had not received such treatment before admission. Each patient was evaluated on admission for signs of neurologic dysfunction. Pre-clinical study: Humanized ACE-2 transgenic knock-in mice received the SARS-CoV-2 spike protein via jugular vein injection for 2 weeks. One group had received Losartan (10 mg/kg), an ARB, in their drinking water for two weeks before the injection, while the other group began Losartan treatment after the spike protein injection. Cognitive functions, cerebral blood flow, and cerebrovascular density were determined in all experimental groups. Moreover, vascular inflammation and cell death were assessed. Results: Signs of neurological dysfunction were observed in 97 out of 177 patients (51%) taking ACEIs/ARBs prior to admission, compared to 32 out of 118 patients (27%) not receiving

**Funding:** This study was supported by American Heart Association 23AIREA1045073 to MA WWW. heart.org The funder did not play any role in the study design, data collection and analysis.

**Competing interests:** Some data were presented as an abstract at the International Stroke Conference 2023 and the MIDYEAR 2023 Clinical Meeting & Exhibition. The authors have declared that no competing interests exist. Funding:This study was supported by American Heart Association 23AIREA1045073 to MA.

ACEI or ARBs. In animal studies, spike protein injection increased vascular inflammation, increased endothelial cell apoptosis, and reduced cerebrovascular density. In parallel, spike protein decreased cerebral blood flow and cognitive function. Our results showed that pre-treatment with Losartan exacerbated these effects. However, post-treatment with Losartan prevented spike protein-induced vascular and neurological dysfunctions. Conclusion: Our clinical data showed that the use of RAS modulators before encountering COVID-19 can initially exacerbate vascular and neurological dysfunctions. Similar findings were demonstrated in the in-vivo experiments; however, the protective effects of targeting the RAS become apparent in the animal model when the treatment is initiated after spike protein injection.

## Introduction

As of November 2023, there have been over 772 million confirmed cases of COVID-19 globally [World Health Organization, 2023]. Initially, COVID-19 was considered a severe acute respiratory syndrome with the potential for fatal pulmonary complications. However, studies have revealed that COVID-19 has a systemic effect impacting various organs and systems in the body, including the central nervous system. During the course of the disease, patients experienced several neurological complications such as headaches, disorientation, brain fog, and attention deficit [1,2]. Cognitive impairment was observed not only in severely affected COVID-19 patients but also in young individuals with mild to moderate disease cases [3]. As the disease is still virulent and threatening the quality of lives of people, greater attention should be focused on the short and long-term impact of COVID-19 on cognitive function and identify possible therapeutic interventions.

Patients with cardiovascular diseases such as diabetes, heart failure, obesity, or hypertension are at a higher risk for COVID-19 due to their advanced age and comorbid conditions [4,5]. Moreover, a significant association exists between hypertension and an increased likelihood of experiencing severe COVID-19 symptoms [6]. RAS modulators, including angiotensin-converting enzyme inhibitors (ACEI) and angiotensin receptor blockers (ARB), are first-line agents for the treatment of heart failure and hypertension. Multiple molecular mechanisms were investigated to explore the cardiovascular protective effects of the RAS modulators. One accepted mechanism is the RAS-induced upregulation of the angiotensin-converting enzyme-2 (ACE-2). ACE-2 degrades Ang II, the bioactive form of RAS, into Ang 1–7, which results in vasodilatory, antioxidant, and anti-inflammatory effects [7]. However, because ACE-2 acts as a binding receptor for SARS-CoV-2, facilitating its cell entry, a general concern was raised about whether RAS modulators will increase the risk of infection and disease progression [8–10].

Moreover, the contribution of RAS modulators to COVID-19-induced cognitive dysfunction is still a gap in knowledge. Therefore, this study aims to investigate the following: 1) Does prior treatment with RAS modulator worsen COVID-19-induced vascular and cognitive dysfunction? 2) Can post-treatment with RAS modulator improve cognitive performance and vascular function following COVID-19? To address these inquiries, a combination of clinical and preclinical research was conducted to shed light on the utility and potential risks associated with the use of ACEI or ARB in the context of COVID-19.

## Methods

### Clinical

The study population for this research was sourced from the St Joseph's/Candler Meditech Electronic Medical Record system. Inclusion criteria were clearly defined as individuals who

had been hospitalized for COVID-19 between 2020 and 2022, and who also had a comorbid condition of hypertension. Data was accessed from May 1, 2023 through June 30, 2023. Authors had access to patient information during the data collection period. These patients were categorized into two distinct cohorts: those who had been receiving ACEI/ARB medications before hospital admission and those who were not on ACEI/ARB therapy prior to their hospitalization. For each patient, we collected baseline demographic information, including age, gender, body mass index, comorbid conditions, concurrent antihypertensive therapy, and average blood pressure on admission. A summary of these baseline characteristics is presented in Table 1.

Our assessment was primarily aimed at identifying indications of neurological dysfunction within the patient cohort. This evaluation encompassed a comprehensive examination of several key parameters, which included assessments for altered mental status, dizziness/vertigo, headache/migraine, loss of coordination, loss of consciousness, muscle weakness, confusion, slurred speech, and visual changes. Additionally, we evaluated patients who had undergone neurologic imaging to ascertain whether there were discernible distinctions between the two cohorts in terms of ischemic changes or atrophy.

## Animals

Humanized ACE-2 Knock-in (ACE-2 KI) mice were purchased from Jackson Laboratory (Jax lab: Stock No: 035800, Ellsworth, Maine, USA) and underwent inbreeding within the animal facility at Mercer University. All animal protocols received approval from the Mercer University Institutional Animal Care and Use Committee (IACUC, accredited by The American

**Table 1. Baseline characteristics of patients admitted to the hospital with a diagnosis of COVID19 and comorbid hypertension.**

| Characteristic | ACEI/ARB prior to admission (n = 177) | No ACEI/ARB prior to admission (n = 118) |
|---|---|---|
| Age | 69 | 70 |
| Sex (male) | 85 (48%) | 43 (43) |
| BMI kg/m$^2$ | 32.8 | 30.9 |
| Race | | |
| Caucasian | 97 (55%) | 68 (58%) |
| African American | 78 (44% | 44 (37%) |
| Other | 2 (1%) | 6 (5%) |
| Concurrent anti-hypertensives | | |
| CCB | 73 (41%) | 52 (44%) |
| Thiazides | 52 (29%) | 26 (22%) |
| Beta blockers | 80 (45%) | 56 (47%) |
| Major Comorbidities<br>Diabetes<br>Chronic kidney disease<br>Dementia<br>Hx of Stroke<br>Hyperlipidemia<br>Chronic heart failure | 83 (43%)<br>22 (12%)<br>9 (5%)<br>17 (10%)<br>84 (47%)<br>19 (10%) | 35 (30%)<br>12 (8%)<br>11 (9%)<br>17 (14%)<br>49 (41%)<br>10 (8%) |
| Average blood pressure on admission | 137/76 | 138/79 |

Abbreviations: Body mass index (BMI, calculated as weight in kilogram divided by the square of height in meters), Angiotensin converting enzyme inhibitor (ACEI), angiotensin receptor blocker (ARB), calcium channel blocker (CCB).

Association for Accreditation of Laboratory Animal Care) and adhered to the current ARRIVE guidelines 2.0. Animals were provided with a standard mouse chow diet and unrestricted tap water access. Mice were kept on a 12-hour light-dark cycle. Animals were sacrificed using carbon dioxide and cervical dislocation. Buprenorphine 0.1mg/kg body weight was injected subcutaneously upon detecting any signs of animal distress.

## Losartan treatment and SARS-CoV-2 spike protein injection

The study involved the administration of the SARS-CoV-2 spike protein through intravenous injection of SARS-CoV-2 nucleoprotein/spike protein recombinant (4ug/animal, Invitrogen, USA, Cat. No. RP-87706) into the jugular vein. Losartan (10 mg/kg body weight, Tokyo Chemical Industry, Tokyo, Japan, Cat. No. L0232) was introduced into the animals' water supply. Losartan treatment commenced either two weeks before the spike protein injection or immediately after the injection. The mice were sacrificed two weeks after receiving the SARS-CoV-2 recombinant spike protein injection. Male and female humanized ACE-2 KI (hACE2 KI) mice were randomly divided blindly into four groups: 1) Control group, 2) SARS-CoV-2 spike protein injection, 3) Pre-Losartan treatment group with SARS-CoV-2 spike protein injection, and 4) SARS-CoV-2 spike protein injection group with post-injection Losartan treatment.

## RT-qPCR

Triazole (Thermo-Fisher, USA, Cat. No. AC345480250) was used to isolate RNA from brain homogenate. The Thermo Scientific NanoDrop 2000C Spectrophotometer (Thermo Scientific, USA) was used to quantify RNA concentrations. The QuantStudio™ 3 Real-Time PCR System (Applied Biosystems, Thermo Scientific, USA) was utilized to run qRT-PCR. Table 2 shows forward and reverse primers used in the study. GAPDH was used for normalization in all experiments.

## Western blot analysis

The expression of ACE-2 and the apoptotic marker, cleaved caspase-3, was detected by western blot. Brain tissues were homogenized in RIPA buffer (Millipore, Billerica, MA, USA, Cat# 3P 20188) and then separated by a 10% SDS-polyacrylamide gel using the Mini PROTEAN Tetra Cell SDS-PAGE Gel electrophoresis kit (Biorad Laboratories Inc, Hercules, CA). Subsequently, the separated proteins were transferred onto nitrocellulose membranes. These membranes were blocked and incubated with primary antibodies overnight (Table 3). The primary antibodies were detected using horseradish peroxidase-conjugated secondary antibody (1:5000. The Western Blots were imaged, and band intensity was quantified using the Azure Biosystems c600 (Azure Biosystems Inc., Dublin, CA) and Image-J software, respectively.

**Table 2. Primers.**

| Gene | Forward | Reverse |
|---|---|---|
| ACE2 | 5'-TCC ATT GGT CTT CTG CCA TCC G-3' | 5'-AGA CCA TCC ACC TCC ACT TCT C-3' |
| TNF-α | 5'-GGT GCC TAT GTC TCA GCC TCT T-3' | 5'-GCC ATA GAA CTG ATG AGA GGG AG-3' |
| Il-6 | 5'-TAC CAC TTC ACA AGT CGG AGG C-3' | 5'-CTG CAA GTG CAT CAT CGT TGT TC-3' |
| GAPDH | 5'-CCA AGA AGT GCT CAG AGA GGT G-3' | 5'-GTC CTT GAA CTT CTT TTT GGT CTC-3' |

**Table 3. Antibodies.**

| Antibody | Vendor | Catalog number |
|---|---|---|
| ACE-2 | Proteintech (Rosemont, IL, USA) | 21115-1-AP |
| B-Actin | R&D (Minneapolis, MN, USA) | MAB8929 |
| Cl. Caspase-3 | R&D (Minneapolis, MN, USA) | MAB835 |

## Vascular density assessment

Brain tissues isolated from the hACE2 KI were fixed and sectioned as described previously by our group [11]. Brain sections were stained with Lycopersicon Esculentum Lectin, DyLight™ 488 (Vector Laboratories, Burlingame, CA, USA, Cat. No. DL-1174-1). Imaging of the sections was performed using a Nikon Eclipse Ti-E Inverted Microscope (Nikon Instruments Inc., Melville, NY). FIJI software was employed to analyze the three-dimensional structures of the Z-stacked images. Vascular density was calculated by dividing the mean density of stained vasculature, as determined by FIJI, by the total number of planes in the Z-stack.

## Cerebral blood flow assessment

Cerebral blood flow was assessed using the RFLSI III Laser Speckle Imaging System (RWD, San Diego, CA, USA). Cerebral blood flow was initially measured at baseline before administering the SARS-CoV-2 spike protein injection, and before sacrifice. Mice were anesthetized using isoflurane to facilitate the procedure, and a vertical incision was made to expose the skull. The surgical site was cleaned and sealed with a clip following the procedure. The percent change in the cerebral blood flow was compared among the groups.

## Cognitive function assessment

Using a Y-shape maze, memory and learning functions were evaluated at baseline and ten days after SARS-CoV-2 recombinant spike protein injection. Briefly, hACE2 KI mice were allowed to explore a Y-maze with only two open arms. On the test, the third arm was open. Using ANY-maze 6.1 tracking software, the time spent in the goal zone (new arm) and the number of entries were calculated. Total distance traveled by each animal is measured to exclude any motor dysfunction that could affect the cognitive functions assessment.

## Statistical analysis

The clinical data analysis was performed employing Excel Data Analysis. Chi-square tests served as the primary statistical method for our analysis. Significance was determined at the conventional threshold of $P < 0.05$, utilizing two-sided testing. Pre-clinical data analysis was conducted using GraphPad Prism version 10.1. For animal studies, the sample size was determined from our previous work. One-way ANOVA was used to assess the differences in the means between control, S-protein Pre-Losartan, and Post-Losartan treatment. Significance was determined at $P < 0.05$. Data is presented as mean ± standard deviation. A Tukey's post-hoc test was used to adjust for the multiple comparisons to assess significant interaction.

## Results

### The impact of receipt of an ACEI/ARB on neurologic dysfunction in patients admitted to the hospital with COVID-19

There were 295 patients included in our study who were admitted to the hospital with COVID-19 and concurrent hypertension. Of those, 177 patients were receiving ACEI/ARBs

prior to their COVID-19 hospitalization, and 118 patients were not on ACEI/ARB therapy before admission. Our findings revealed a striking difference in the occurrence of neurologic dysfunction between the two groups. Among the patients who were taking ACEI/ARB medications prior to hospitalization, 91 individuals (51.4%) exhibited signs of neurologic dysfunction in contrast to 32 patients (27.1%) who were not on ACEI/ARB therapy before admission (P < 0.001) as reported in Table 4.

Among the patients experiencing neurological dysfunction, 42 individuals underwent neurologic imaging through either a magnetic resonance imaging (MRI) or computerized tomography (CT) scan, with 27 patients belonging to the ACEI/ARB group and 15 patients in the no ACEI/ARB group. Notably, 66.7% (18 out of 27) of the patients on ACEI/ARB therapy prior to admission displayed ischemic changes or atrophy on their neurologic imaging. In comparison, 53.3% (8 out of 15) of patients who were not on ACEI/ARB therapy before admission exhibited similar neuroimaging findings.

## The impact of losartan treatment on ACE-2 expression

hACE2 KI mice received Losartan treatment for 2 weeks, other groups were intravenously injected with recombinant SARS-CoV-2 spike protein with or without Losartan treatment. To evaluate the impact of Losartan and spike protein on ACE-2 expression, we assessed ACE-2 gene and protein expression in brain homogenates using real-time PCR and Western blot analysis. The group that received Losartan only displayed an increase in ACE-2 expression. However, spike protein injection significantly reduced the ACE-2 gene and protein expressions, which were prevented by the post-Losartan treatment. (Fig 1A and 1B).

## The impact of pre and post-losartan treatment on inflammation and cell death

hACE2 KI mice were intravenously injected with recombinant SARS-CoV-2 spike protein. Losartan treatment began either two weeks before the spike protein injection or immediately after the injection. The mice were sacrificed two weeks after receiving the SARS-CoV-2 recombinant spike protein injection. We first evaluated the effect of SARS-CoV-2 spike protein on brain inflammation. RNA was isolated from brain homogenate and examined for inflammatory markers. Our results show that SARS-CoV-2 spike protein increased gene expression of inflammatory markers such as TNF-alpha and IL-6. (Fig 2A). Post-losartan treatment was

**Table 4. The primary type of neurologic dysfunction in patients who were on ACEI/ARB prior to admission in contrast to the non-ACEI/ARB treated group.**

| Type of Neurological Dysfunction | ACEI/ARB prior to admission (n = 91) | No ACEI/ARB prior to admission (n = 32) |
|---|---|---|
| Altered mental status | 23 | 8 |
| Dizziness/vertigo | 7 | 3 |
| Headache/migraine | 9 | 2 |
| Loss of coordination | 4 | 0 |
| Loss of consciousness | 3 | 0 |
| Muscle weakness | 37 | 15 |
| Confusion | 4 | 4 |
| Slurred speech | 3 | 0 |
| Visual changes | 1 | 0 |

Abbreviations: Angiotensin-converting enzyme inhibitor (ACEI), angiotensin receptor blocker (ARB).

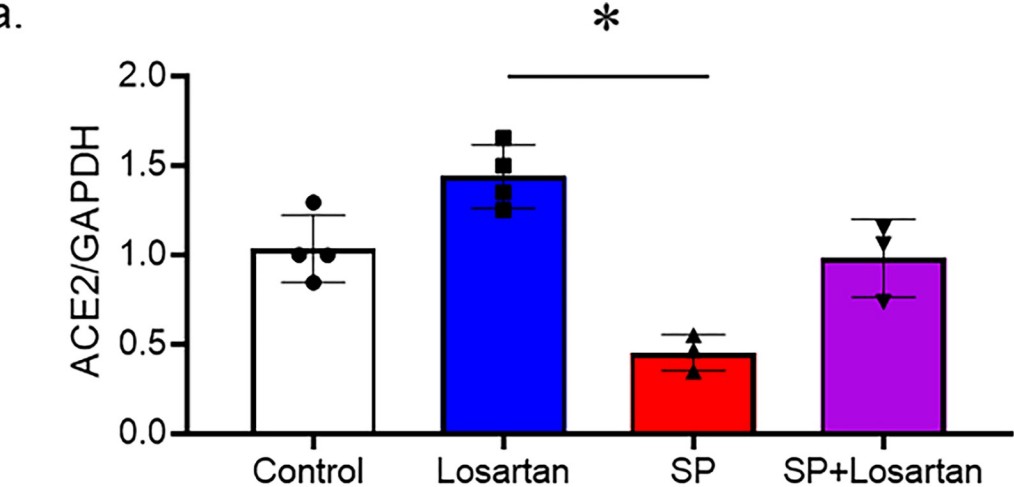

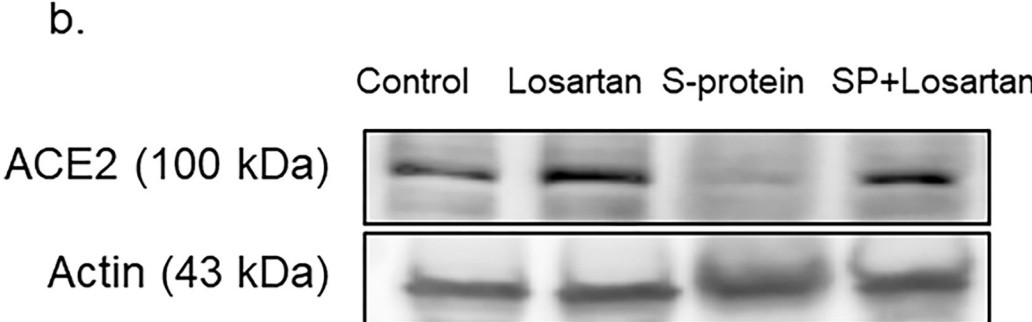

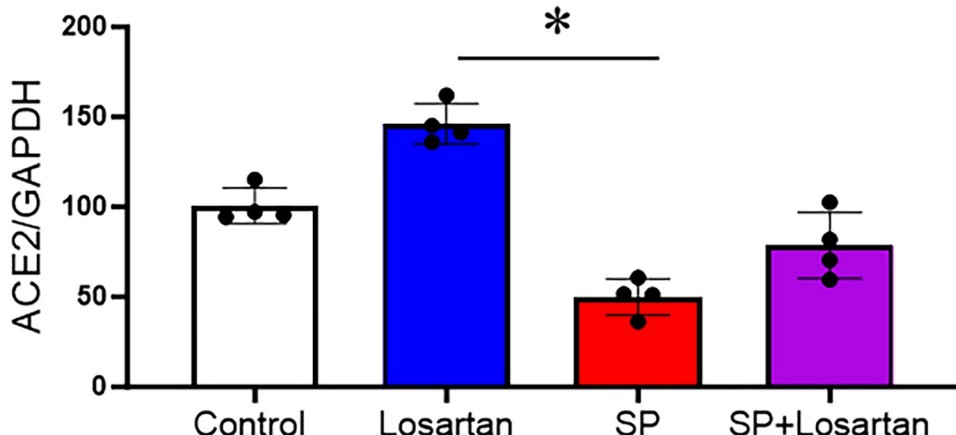

**Fig 1. Losartan increased ACE-2 expression, while SARS-CoV-2 spike protein decreased ACE-2 expression in hACE2 brains.** hACE2 KI mice were treated only with Losartan or intravenously injected with SARS-CoV-2 spike protein (SP, 4 μg/animal) with or without Losartan (10 mg/kg body weight) for 2 weeks. Brain homogenate was assessed for ACE-2 expression. a) RT-PCR analysis showing the effect of Losartan and spike protein on ACE-2 gene expression. Losartan treatment increased ACE-2 gene expression, while spike protein significantly reduced ACE-2 gene expression. b) Western blot analysis for ACE-2

expression in brain homogenate. Our results showed that Losartan treatment increased ACE-2 expression while spike protein decreased ACE2 expression. Treatment of Losartan after spike protein restored ACE-2 expression in hACE2 mice treated with spike protein. (One-way ANOVA, *P<0.05, n = 3–4).

more effective in reducing inflammatory markers compared to pre-losartan treatment. To evaluate the impact of pre and post-Losartan treatment on cell death, cleaved caspase-3 expression in brain homogenates was assessed using western blot. The expression of cleaved caspase-3, an apoptotic marker, was significantly enhanced following the spike protein injection. However, only post-Losartan treatment prevented SARS-CoV-2 spike protein-induced cell death (Fig 2B).

### The impact of pre and post-losartan treatment on vascular density

Our results showed that SARS-CoV-2 spike protein injection caused a vascular rarefaction as seen by decreased vascular density compared to control. In agreement with the immunoblotting results, only post-Losartan treatment prevented the reduction in vascular density. Interestingly, the decrease in vascular density was further enhanced in the mice who received Losartan two weeks before SARS-CoV-2 spike protein injection (Fig 3A and 3B).

### The impact of pre and post-losartan treatment on cerebral blood flow

hACE2 KI started the Losartan treatment either two weeks before receiving the intravenous injection of the recombinant SARS-CoV-2 spike protein or immediately after the injection. The impact of the pre/post Losartan treatment on cerebral blood flow was assessed using laser speckle imaging at baseline and before sacrifice. Our results showed that pre-Losartan treatment caused a reduction in the cerebral blood flow, which was significant compared to the control (*P<0.05), and marginally non-significant compared to the spike protein group (P = 0.052). On the other hand, the percent change in the cerebral blood was similar in both the control and post-Losartan treatment groups (Fig 4A and 4B).

### The impact of pre and post-losartan treatment on cognitive function

The Y-maze was used to assess short-term memory in hACE2 mice who received intravenous injections of SARS-CoV-2 spike protein. We also evaluated the different start times of Losartan treatment. Our results showed that SARS-CoV-2 spike protein injection caused a reduction in the time spent in the novel arm compared to the control. This reduction in time was further pronounced in the group of mice that had initiated Losartan treatment two weeks prior to SARS-CoV-2 spike protein injection. However, post-Losartan treatment prevented spike protein-induced cognitive dysfunction, as evidenced by the increased amount of time the mice spent exploring the arm they hadn't visited before. Furthermore, we determined the total distance traveled by each animal to exclude any motor disability that could affect Y-maze results. We did not detect any motor disability due to losartan or spike protein injection (Fig 5A and 5B).

## Discussion

The current study showed that individuals receiving ACEI/ARB therapy prior to their COVID-19 hospitalization experienced a significantly higher risk of experiencing neurologic dysfunction compared to those not on such therapy. In agreement, our animal studies showed that SARS-CoV-2 spike protein with prior treatment with Losartan, an ARB, had increased cerebrovascular inflammation and cell death, which were coupled with cognitive dysfunction.

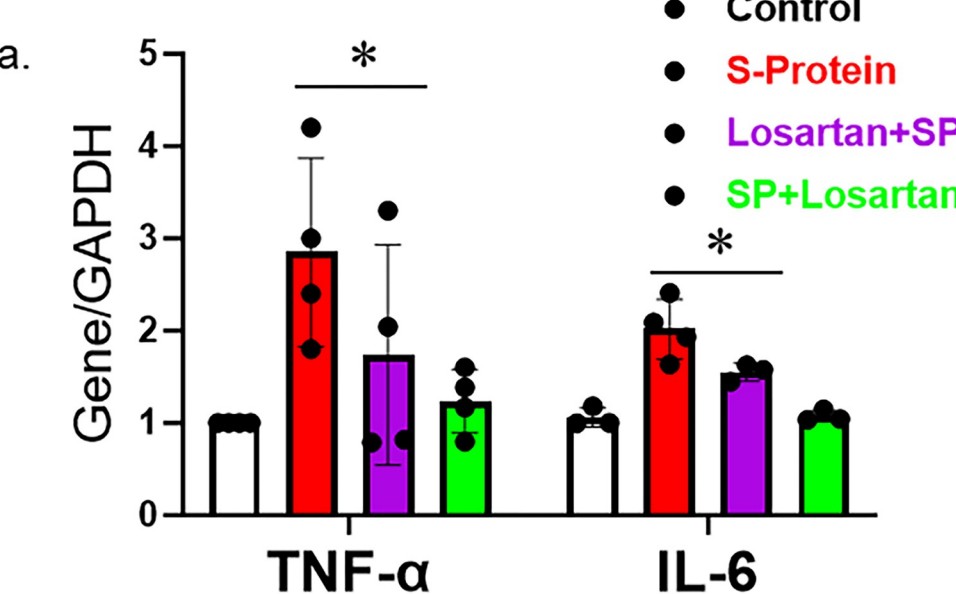

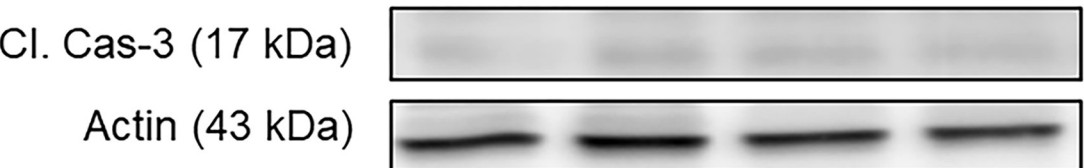

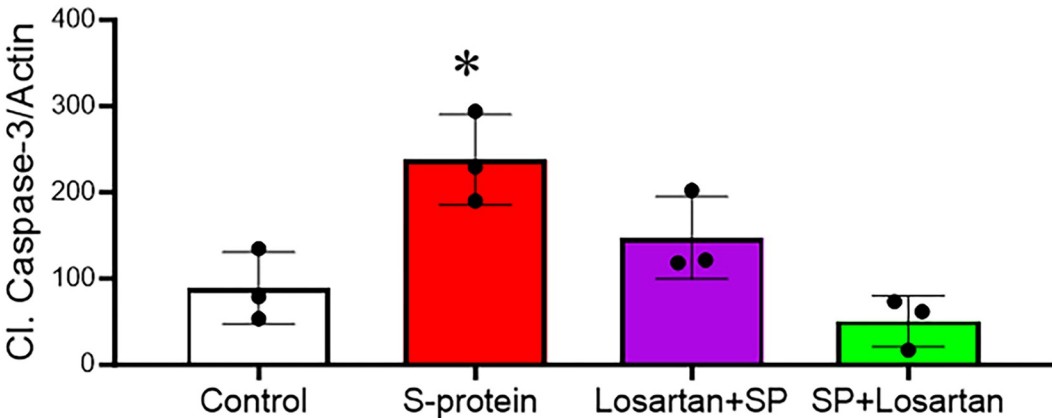

**Fig 2. Only post-losartan treatment prevented SARS-CoV-2 spike protein-induced inflammation/apoptosis but not pre-losartan treatment.** hACE2 KI mice were injected with recombinant SARS-CoV-2 spike protein (4 mg/animal). Losartan treatment (10 mg/kg) began either two weeks before the spike protein injection (Losartan + SP) or immediately after the injection (SP + Losartan). Whole brain homogenate was assessed for inflammatory markers and apoptotic markers. a) RT-PCR analysis of inflammatory markers (TNF-α and Il-6) in hACE-2 brains. Our results show that spike protein injection caused a significant increase in TNF-α and Il-6 gene expression. Pre-

Losartan treatment showed a similar increase in TNF-α and Il-6 gene expression. Post-Losartan treatment inhibited spike protein-induced increased inflammation. b) Western Blot analysis for apoptotic marker, cleaved caspase-3. Our results showed that spike protein and pre-losartan treatment increased caspase activation, while post-losartan treatment showed a protective effect against spike protein-induced cell death. (One-way ANOVA, *$P<0.05$, n = 4).

Remarkably, post-treatment with Losartan displayed a protective effect, preventing COVID-19-induced cerebrovascular and cognitive dysfunction. Our study underscores the importance of the de novo initiation of RAS modulators following COVID-19 infection.

The renin-angiotensin system plays a dual role in our body, exhibiting both protective and harmful effects depending on the abundance of angiotensin II and the activation of Angiotensin receptors [12,13]. Ang II binds with two main receptors, $AT_1R$ and $AT_2R$, and induce different physiological responses. The augmented activation of $AT_1R$ by Ang II constitutes the detrimental arm of RAS that causes inflammation, apoptosis, and oxidative stress [14,15]. However, $AT_2R$ activation contributes to more vascular protective effects such as vasodilation and anti-inflammatory effects [16]. Moreover, ACE-2 is an enzyme that catalyzes the degradation of Ang II to Ang-(1–7), and the resultant molecules activate the Mas receptor, which results in protective actions [17–19]. Therefore, Ang-(1–7)/Mas axis and the $AT_2R$ constitute the protective arm of the RAS. In COVID patients, studies have shown a shift in balance towards the RAS harmful arm due to the downregulation of ACE2 and the overactivation of $AT_1R$ by angiotensin II [20,21]. Therefore, using a RAS modulator as ACEI or ARB is a logical approach to restore the balance and enhance the protective arm of RAS. Our group has recently shown that using Losartan restored the RAS balance and reduced SARS-CoV-2-induced cerebrovascular dysfunction [22]. On the other side, ACE-2 also serves as a receptor that facilitates the entrance of SARS-COV-2 spike protein into the host cells [23]. Notably, experimental studies have shown that RAS modulators could upregulate ACE-2 expression [24–26]. This has raised several concerns regarding whether pretreatment with RAS modulators could worsen COVID-19 outcomes.

Since the emergence of COVID-19, various studies have investigated factors linked to heightened disease severity, including the effects of ACEI/ARB medications. In 2020, Flacco et al. conducted a meta-analysis aiming to determine any potential connection between ACEIs or ARBs and severe or fatal COVID-19 outcomes. Their findings revealed no substantial correlation between the usage of these medications and an increased risk of severe or lethal outcomes in COVID-19 patients [27]. In 2021, Grover and Oberoi conducted a systematic review and meta-analysis with the objective of evaluating the clinical consequences for COVID-19 patients using ACEIs or ARBs. The study involved a comprehensive analysis of various clinical indicators among patients on these medications. The authors' conclusion was that the utilization of ACEIs or ARBs did not significantly impact the clinical outcomes including severity of disease and mortality in patients with COVID-19 [28]. Also, in 2021, Singh et al. conducted a systematic review, meta-analysis, and meta-regression analysis to explore the connection between ACEIs, ARBs, and the severity and mortality of COVID-19 patients. Their study meticulously examined various datasets and research. The authors reached the conclusion that the utilization of ACEIs or ARBs did not lead to a significant increase in the severity or mortality of COVID-19 among patients [29]. In 2022, Gnanenthiran et al. published a meta-analysis to assess the safety and efficacy of RAS inhibition in adults with COVID-19. The authors found no significant difference in all-cause mortality between patients receiving RAS inhibition therapy and those without it. However, there was a borderline decrease in acute myocardial infarction among patients on RAS inhibitors. On the other hand, there was an increased risk of acute kidney injury (AKI) associated with RAS inhibitors use, particularly in

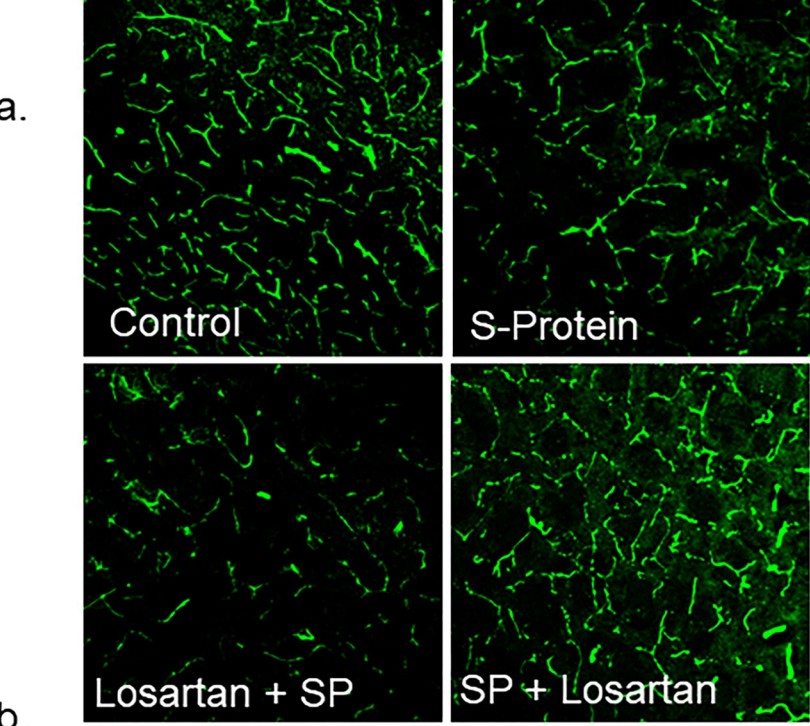

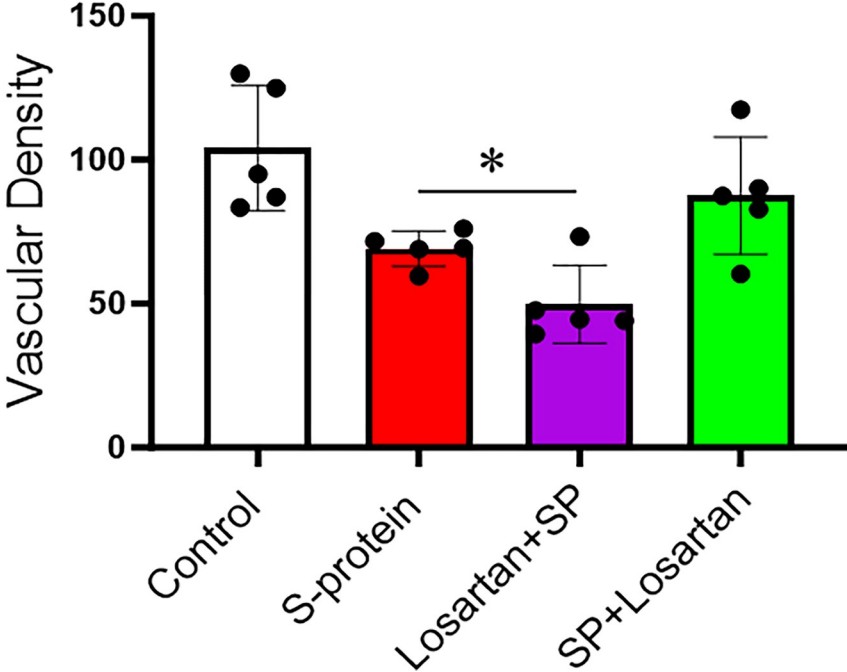

**Fig 3. Post-losartan treatment prevented SARS-CoV-2 Spike protein-induced vascular rarefaction but not pre-losartan treatment.** hACE2 KI mice were injected with recombinant SARS-CoV-2 spike protein (4 mg/animal). Losartan treatment (10 mg/kg) began either two weeks before the spike protein injection (Losartan + SP) or immediately after the injection (SP + Losartan). Brains were isolated and sectioned 30–40 um. The brain section was stained for vasculature. 3D confocal images were reconstructed to assess vascular density using FIJI software. a)

Representative images of the brain cortex. b) Image analysis showing that SARS-CoV-2 caused vascular rarefaction and significantly reduced vascular density. Pre-Losartan treatment showed decreased vascular density. Only post-losartan treatment showed a protective effect and restored vascular density compared to control. (One-way ANOVA, $^*P<0.05$, n = 5).

hospitalized patients. Despite this increased risk of AKI, there was no increase in the need for dialysis or other adverse outcomes such as congestive cardiac failure, stroke, or venous thromboembolism [30]. Most of the studies have focused on clinical outcomes, such as mortality and hospitalization duration. Given the intricate neurological implications of COVID-19, assessing the impact of ACEI/ARBs in these patients is of great importance, especially considering the well-established benefits of these medications in managing numerous chronic health conditions. In the clinical arm of our study, we assessed the correlation between the administration of a RAS modulators and the signs of neurological dysfunction, including but not limited to headache, altered mental status, dizziness, and loss of consciousness. A significant disparity was noted in the occurrence of neurologic dysfunction between the groups, with patients receiving ACEI/ARBs medication prior to admission showing more signs of neurologic dysfunction compared to those not on an ACEI/ARBs prior to admission. While these findings suggest a potential adverse effect of ACEI/ARBs, given their beneficial effects on various medical conditions, we aimed to delve deeper into the outcomes for patients who maintained ACEI/ARBs usage during and after COVID-19, as well as those who started ACEI/ARBs therapy around the same time as the diagnosis of COVID-19. Unfortunately, the data was not readily accessible in the electronic medical records.

To address the lack of a post-treatment arm in the clinical study, our research group used one of the well-established COVID-19 animal models [22,31] to test the impact of timing when administering RAS modulators. The in-vivo data showed that prior treatment with Losartan, an ARB, was associated with an upregulation in inflammation and cell death, in addition to a reduction in vascular density and cerebral blood flow. Those vascular and molecular changes were coupled with an increase in cognitive dysfunction. However, the de novo initiation of a Losartan subsequent to spike protein injection effectively prevented COVID-19-induced inflammation, apoptosis, and cognitive dysfunction. A plausible explanation lies in the ability of Losartan to block the $AT_1$ receptor, thereby inhibiting the harmful arm and eventually restoring a balanced RAS equilibrium. Additionally, unopposed $AT_2R$ stimulation could be a mechanism underlying the protective effect of Losartan demonstrated in the study. In this study, we are proposing that timing matters when we administer a RAS modulator.

In the clinical arm of our study, we investigated the correlation between the administration of a RAS modulator and the presence of neurological dysfunction, encompassing symptoms such as headache, altered mental status, dizziness, and loss of consciousness. Our analysis revealed a notable discrepancy in the occurrence of neurological dysfunction between patient groups, with those receiving ACE/ARB medication prior to admission exhibiting a higher prevalence of neurological symptoms compared to individuals not on ACE/ARB therapy before hospitalization.

However, it is crucial to acknowledge the inherent limitations of the clinical aspect of our investigation. Firstly, the retrospective nature of our study introduces potential selection bias and may restrict access to comprehensive clinical data for all patients. Additionally, the lack of uniform neurological testing, such as CT scans or MRIs, for all individuals could impact the comprehensiveness of our findings. While we attempted to address this limitation by analyzing available clinical data and documenting subjective complaints, the variability in diagnostic evaluations remains a notable constraint. Moreover, the modest sample size and the

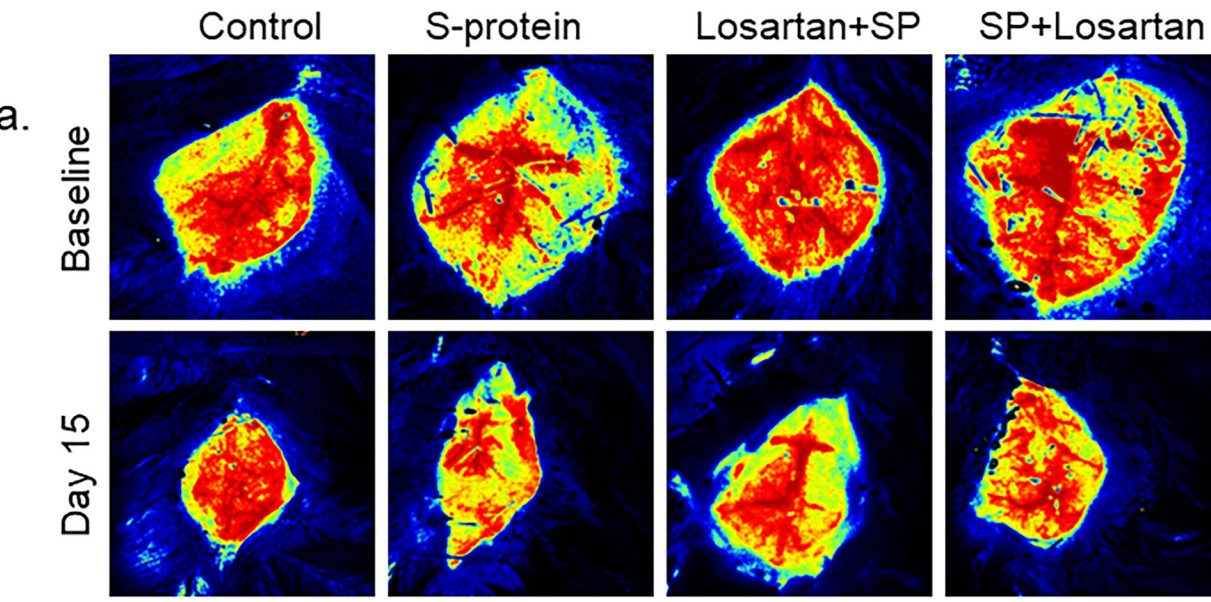

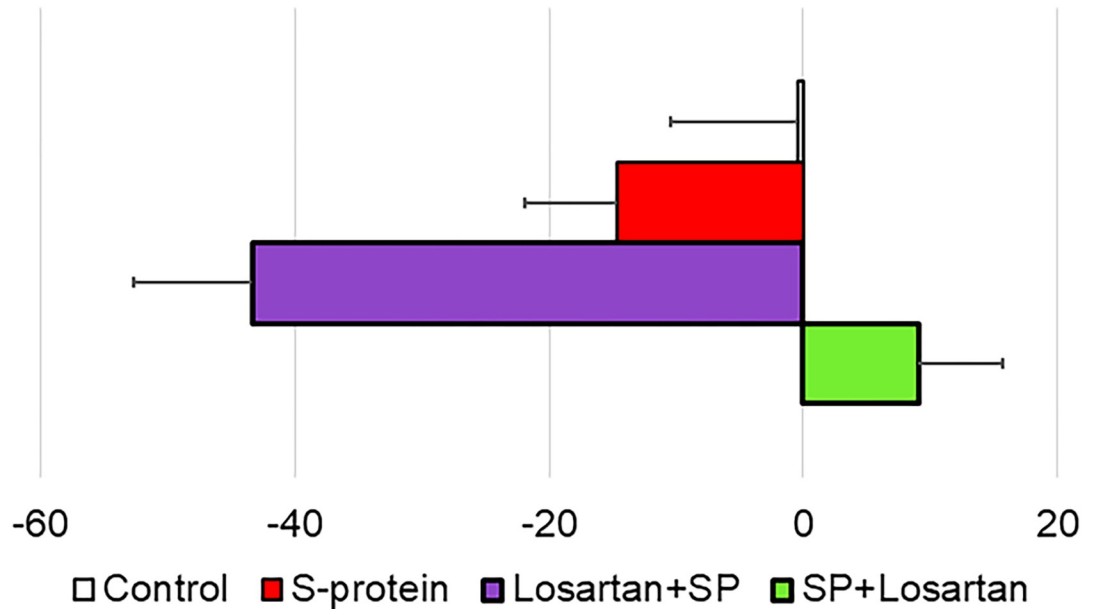

**Fig 4. Pre-losartan treatment reduced cerebral blood flow after SARS-CoV-2 spike injection but not post-losartan treatment.** hACE2 KI mice were injected with recombinant SARS-CoV-2 spike protein (4 mg/animal). Losartan treatment (10 mg/kg) began either two weeks before the spike protein injection (Losartan + SP) or immediately after the injection (SP + Losartan). a) Representative images of cerebral blood flow were measured at baseline and 2 weeks after spike protein injection. b) Percent change in cerebral blood flow after 15 days compared to baseline. Our results showed cerebral blood flow was reduced in pre-Losartan treatment group compared to the control (*$P < 0.05$). Post-Losartan treatment group was similar to the control. (One-way ANOVA, $P < 0.05$, n = 3–7).

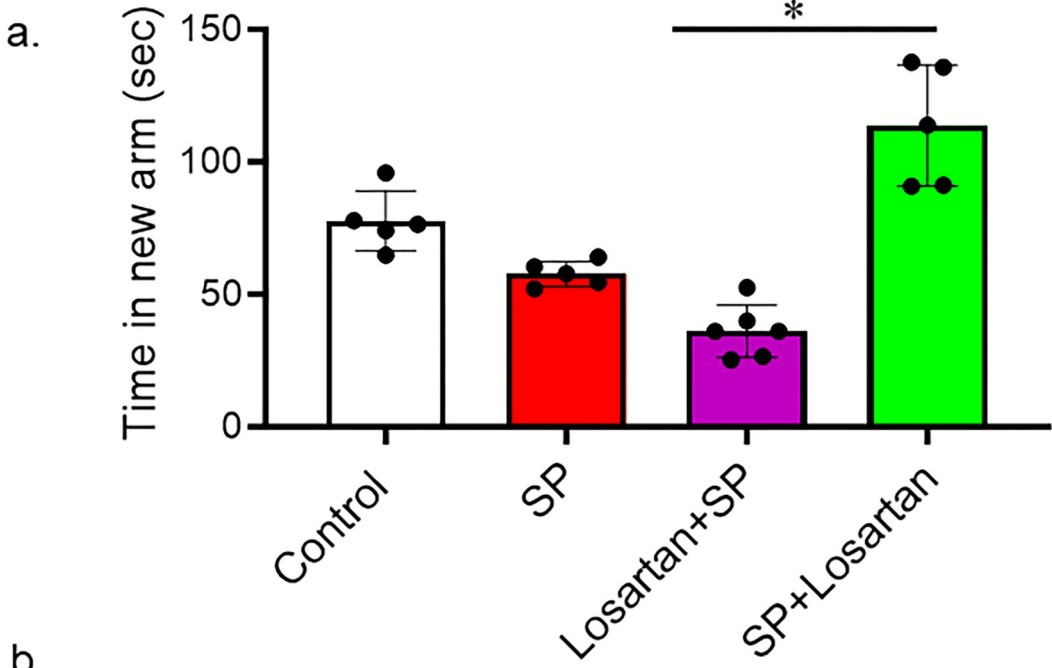

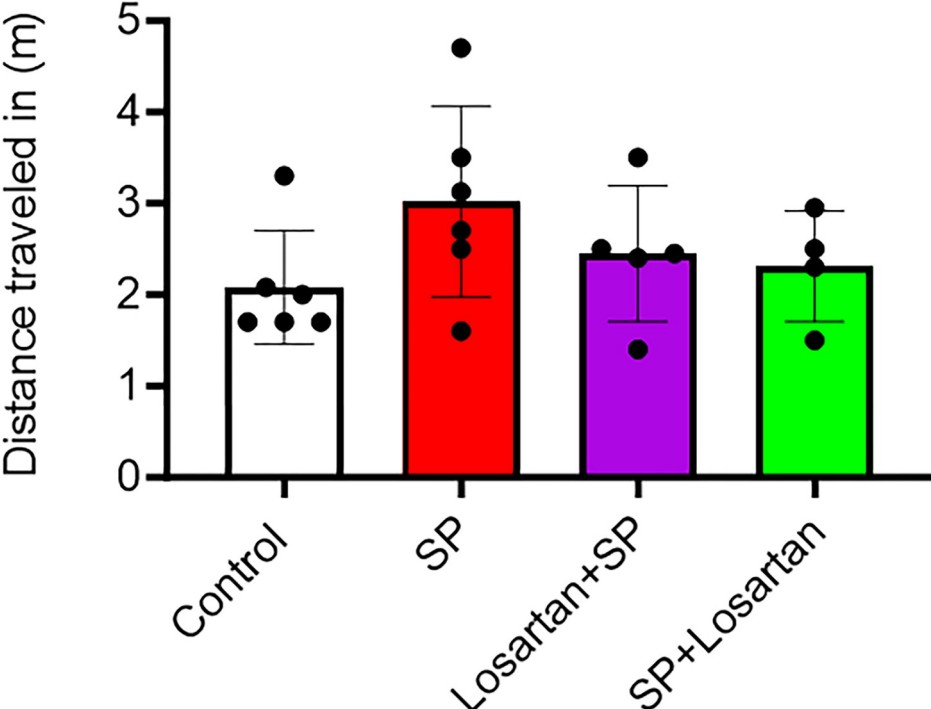

**Fig 5. Post-losartan treatment prevented SARS-CoV-2 spike protein-induced decreased cognitive impairment but not pre-losartan treatment.** a) Learning and memory functions were assessed using Y-maze. Spike protein and pre-losartan treatment showed significant impairment in cognitive function, as seen by reduced time spent in the new arm. Only post-losartan treatment showed improvement in cognitive functions after spike protein injection. b) The total distance traveled was measured to detect any motor disability with spike protein injection. No significant results were observed in the total distance traveled between groups. (One-way ANOVA, *P<0.05, n = 5).

disproportionate representation of African American patients in our study population may limit the generalizability of our results. Despite our efforts to contextualize our findings within the demographic framework of our study location, it is important to acknowledge the potential influence of demographic factors on our conclusions.

Regardless of these limitations, our study underscores the significance of recognizing neurological complications in COVID-19 patients undergoing RAAS inhibition, particularly considering the established therapeutic benefits of these medications in managing chronic health conditions. While our findings suggest a potential association between ACE/ARB use and neurological symptoms, further investigation into the complex relationship between ACE/ARB medications and neurological manifestations in COVID-19 patients is warranted.

In conclusion, this study contributes significant insights into the complex relationship between RAS modulators and COVID-19-induced cognitive and vascular dysfunction. The findings suggest that the timing of RAS modulator treatment plays a critical role in its effectiveness in mitigating cognitive and vascular damage in COVID-19. Future studies should address the limitation of our preclinical model, which exclusively utilized ARBs therapy, by incorporating ACEI to mimic the clinical arm where patients received either ARBs or ACEI. Additionally, further studies are needed to assess the reproducibility of the experimental data in a clinical setting. It is also essential to test whether the continuation of RAS modulators after COVID-19 could reverse the observed vascular and cognitive dysfunction. The current study highlights the importance of further investigation into the relationship between ACEI/ARBs medications and neurologic symptoms in COVID-19 patients, especially due to the profound benefits of these medications in multiple disease states.

## Supporting information

**S1 Raw images. Original blots.**
(PDF)

## Author Contributions

**Conceptualization:** Mackenzi Meier, Mohammed Abdelsaid, Maha Coucha.

**Data curation:** Mackenzi Meier, Sara Becker, Erica Levine, Oriana DuFresne, Kaleigh Foster, Joshua Moore, Faith N. Burnett, Veronica C. Hermanns, Stan P. Heath, Mohammed Abdelsaid, Maha Coucha.

**Formal analysis:** Mackenzi Meier, Kaleigh Foster, Faith N. Burnett, Mohammed Abdelsaid, Maha Coucha.

**Funding acquisition:** Mohammed Abdelsaid.

**Investigation:** Mackenzi Meier, Mohammed Abdelsaid, Maha Coucha.

**Methodology:** Mackenzi Meier, Sara Becker, Erica Levine, Oriana DuFresne, Kaleigh Foster, Joshua Moore, Faith N. Burnett, Veronica C. Hermanns, Stan P. Heath, Mohammed Abdelsaid, Maha Coucha.

**Project administration:** Mackenzi Meier, Sara Becker, Erica Levine, Oriana DuFresne, Kaleigh Foster, Joshua Moore, Faith N. Burnett, Veronica C. Hermanns, Stan P. Heath, Mohammed Abdelsaid.

**Resources:** Mackenzi Meier, Mohammed Abdelsaid.

**Software:** Mohammed Abdelsaid.

**Supervision:** Mackenzi Meier, Mohammed Abdelsaid, Maha Coucha.

**Writing – original draft:** Mackenzi Meier, Mohammed Abdelsaid, Maha Coucha.

**Writing – review & editing:** Mackenzi Meier, Sara Becker, Erica Levine, Oriana DuFresne, Kaleigh Foster, Joshua Moore, Faith N. Burnett, Veronica C. Hermanns, Stan P. Heath, Mohammed Abdelsaid, Maha Coucha.

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
