## [Decision Letter · Decision Letter 0]

29 Feb 2024

PONE-D-24-03599Timing Matters in the Use of Renin-Angiotensin System Modulators and COVID-Related Cognitive and Cerebrovascular DysfunctionPLOS ONE

Dear Dr. Coucha,

Thank you for submitting your manuscript to PLOS ONE. After careful consideration, we feel that it has merit but does not fully meet PLOS ONE’s publication criteria as it currently stands. Therefore, we invite you to submit a revised version of the manuscript that addresses the points raised during the review process.

We look forward to receiving your revised manuscript.

Kind regards,

Michael Bader

Academic Editor

PLOS ONE

Journal Requirements:

2. To comply with PLOS ONE submissions requirements, in your Methods section, please provide additional information regarding the experiments involving animals and ensure you have included details on (1) methods of sacrifice, and (2) efforts to alleviate suffering.

"Some data were presented as an abstract at the International Stroke Conference 2023 and the MIDYEAR 2023 Clinical Meeting & Exhibition.

The authors have declared that no competing interests exist.

Funding:This study was supported by American Heart Association 23AIREA1045073 to MA."

Please confirm that this does not alter your adherence to all PLOS ONE policies on sharing data and materials, by including the following statement: ""This does not alter our adherence to  PLOS ONE policies on sharing data and materials.” (as detailed online in our guide for authors http://journals.plos.org/plosone/s/competing-interests).  

If there are restrictions on sharing of data and/or materials, please state these. Please note that we cannot proceed with consideration of your article until this information has been declared. 

5. In the online submission form you indicate that your data is not available for proprietary reasons and have provided a contact point for accessing this data. Please note that your current contact point is a co-author on this manuscript. According to our Data Policy, the contact point must not be an author on the manuscript and must be an institutional contact, ideally not an individual. Please revise your data statement to a non-author institutional point of contact, such as a data access or ethics committee, and send this to us via return email. Please also include contact information for the third party organization, and please include the full citation of where the data can be found.

7. We note that you have included the phrase “data not shown” in your manuscript. Unfortunately, this does not meet our data sharing requirements. PLOS does not permit references to inaccessible data. We require that authors provide all relevant data within the paper, Supporting Information files, or in an acceptable, public repository. Please add a citation to support this phrase or upload the data that corresponds with these findings to a stable repository (such as Figshare or Dryad) and provide and URLs, DOIs, or accession numbers that may be used to access these data. Or, if the data are not a core part of the research being presented in your study, we ask that you remove the phrase that refers to these data.

Reviewers' comments:

Reviewer's Responses to Questions

**Comments to the Author**

1. Is the manuscript technically sound, and do the data support the conclusions?

Reviewer #1: No

Reviewer #2: Yes

Reviewer #3: No

2. Has the statistical analysis been performed appropriately and rigorously? 

Reviewer #1: No

Reviewer #2: Yes

Reviewer #3: I Don't Know

3. Have the authors made all data underlying the findings in their manuscript fully available?

Reviewer #1: No

Reviewer #2: No

Reviewer #3: No

4. Is the manuscript presented in an intelligible fashion and written in standard English?

Reviewer #1: Yes

Reviewer #2: Yes

Reviewer #3: Yes

5. Review Comments to the Author

Reviewer #1: 1. The authors must describe a technically sound piece of scientific research with data that supports the conclusions. Experiments must have been conducted rigorously, with appropriate controls, replication, and sample sizes

2. The figures lack legends that describe the statistical analyzes and sample sizes of each of the experimental groups.

3. The authors must validate that the preclinical model used emulates the recruited patients. In the preclinical model it is not specified whether they used only males or both males and females. The preclinical model used by the authors. Validation of the preclinical model is missing.

4. The patients received ACEI/ARB, however the mice received only ARB, losartan. The treatment with ACEi remains to be incorporated in the preclinical model.

Reviewer #2: The paper is well written and addresses an important question of the timing of RAS modulators in COVID-19 related cognitive dysfunction. They show statistically better outcomes in patients receiving the medication post infection. I think the paper is acceptable in its current form.

Reviewer #3: In this clinical/preclinical paper, the authors make an attempt to correlate RAS-inhibition before or post onset of Covid-19 with neurological dysfunction. While the preclinical part provides interesting results, appears to be well done and methodologically sound, I have major concerns with respect to the clinical part:

In general, the observed increase in neurological dysfunction in patients on RAS inhibition my be simply due to the fact that these patients were more severely ill, had higher BP levels to start with or cardio-renal problems etc. which afforded additional treatment with RAS inhibitors.

- The number of patients is relatively small. Meaningful statistics are difficult with such small numbers.

- The proportion of black patients is too high to be representative for the US population, let alone the rest of the world.

- Data on comorbidities ar lacking.

- No information on BP control

- No information on the methods to evaluate mental status

- Coincidence of neurological problems: The number (91) with neurological dysfunction in RAS inhibitor- treated patients does not distinguish between i) one dysfunction in one patient or ii) several neurological problems in one patient.

Minor problems:

- line 318: Ang II has the same affinity to AT1R and AT2R. The difference comes rather from the number of receptors in a given tissue under given circumstances.

- line 320: The AT2R-axis of the "protective RAS" needs to be referenced, e.g. Steckelings et al. Pharm. Reviews 2022.

- line 323: The protective RAS is not only constituted by the Ang 1-7/Mas axis but also via the AT2R. Stimulation of the unopposed AT2R could be a mechanism of the protective effects of Losartan post SP in the animals.

- line 348: Meta-analysis on Covid-19/ RAS inhibition by Gnanenthiran SR et al., J Am Heart Assoc. 2022

should be included.

6. PLOS authors have the option to publish the peer review history of their article (what does this mean?). If published, this will include your full peer review and any attached files.

Reviewer #1: No

Reviewer #2: **Yes: **Abhinav Grover

Reviewer #3: No

---

## [Author Response · Author response to Decision Letter 0]

2 Apr 2024

Response to Journal Requirements:

The file naming was adjusted to follow PLOS ONE’s style requirements

2. To comply with PLOS ONE submissions requirements, in your Methods section, please provide additional information regarding the experiments involving animals and ensure you have included details on (1) methods of sacrifice, and (2) efforts to alleviate suffering.

We have updated the method section as directed with the following: “Animals were sacrificed using carbon dioxide and cervical dislocation. Buprenorphine 0.1 mg/kg body weight was injected subcutaneously upon detecting any signs of animal distress.”

We have updated the funding information.

"Some data were presented as an abstract at the International Stroke Conference 2023 and the MIDYEAR 2023 Clinical Meeting & Exhibition.

The authors have declared that no competing interests exist.

Funding:This study was supported by American Heart Association 23AIREA1045073 to MA."

Please confirm that this does not alter your adherence to all PLOS ONE policies on sharing data and materials, by including the following statement: "This does not alter our adherence to PLOS ONE policies on sharing data and materials.” (as detailed online in our guide for authors http://journals.plos.org/plosone/s/competing-interests). 

If there are restrictions on sharing of data and/or materials, please state these. Please note that we cannot proceed with consideration of your article until this information has been declared. 

“This does not alter our adherence to PLOS ONE policies on sharing data and materials.” was added to the manuscript and the cover letter as directed

5. In the online submission form you indicate that your data is not available for proprietary reasons and have provided a contact point for accessing this data. Please note that your current contact point is a co-author on this manuscript. According to our Data Policy, the contact point must not be an author on the manuscript and must be an institutional contact, ideally not an individual. Please revise your data statement to a non-author institutional point of contact, such as a data access or ethics committee, and send this to us via return email. Please also include contact information for the third party organization, and please include the full citation of where the data can be found.

All relevant data are within the manuscript and its Supporting Information files. The online submission was updated accordingly

The original uncropped images are submitted.

7. We note that you have included the phrase “data not shown” in your manuscript. Unfortunately, this does not meet our data sharing requirements. PLOS does not permit references to inaccessible data. We require that authors provide all relevant data within the paper, Supporting Information files, or in an acceptable, public repository. Please add a citation to support this phrase or upload the data that corresponds with these findings to a stable repository (such as Figshare or Dryad) and provide and URLs, DOIs, or accession numbers that may be used to access these data. Or, if the data are not a core part of the research being presented in your study, we ask that you remove the phrase that refers to these data.

The phrase was removed.

Response to Reviewers

Reviewer #1: 

1. The authors must describe a technically sound piece of scientific research with data that supports the conclusions. Experiments must have been conducted rigorously, with appropriate controls, replication, and sample sizes 

We thank the reviewer for their valuable comment. We assure the reviewer that we have adopted the NIH rigor policy and that we comply with transparent standards in our research. All our experiments have been rigorously designed. Our method section describes all experimental procedures in detail for transparency and reproducibility. Proper controls and animal randomization have been used to reduce bias. Our sample size was selected to ensure a power analysis at a significance level of P<0.05. Our animal studies used male and female h ACE-2 mice to avoid biological variables. Finally, all our antibodies and chemical reagent sources have been stated in our manuscript. These reagents have been authenticated by their vendor. 

2. The figures lack legends that describe the statistical analyzes and sample sizes of each of the experimental groups. 

We apologize for this misunderstanding. According to the PLOS One manuscript author guide, all the figure legends are included in the manuscript body and they include sample size and statistical method, as directed. 

3. The authors must validate that the preclinical model used emulates the recruited patients. In the preclinical model, it is not specified whether they used only males or both males and females. The preclinical model used by the authors. Validation of the preclinical model is missing.

We apologize for the mistake and totally agree with the reviewer that the pre-clinical model must be validated. We and others have used hACE-2 mice in COVID-19 research. We have included the reference for studies using the animal model in our discussion. Both male and female hACE-2 mice have been used in our experimental design to avoid biological variables. We have updated our method section accordingly. 

(1) Munoz-Fontela C, Dowling WE, Funnell SGP, Gsell PS, Riveros-Balta AX, Albrecht RA, et al. Animal models for COVID-19. Nature. 2020;586(7830):509-15.

(2) Burnett FN, Coucha M, Bolduc DR, Hermanns VC, Heath SP, Abdelghani M, et al. SARS-CoV-2 Spike Protein Intensifies Cerebrovascular Complications in Diabetic hACE2 Mice through RAAS and TLR Signaling Activation. Int J Mol Sci. 2023;24(22).

 4. The patients received ACEI/ARB, however the mice received only ARB, losartan. The treatment with ACEi remains to be incorporated in the preclinical model. 

We thank the reviewer for their constructive feedback. While patients received ACEI or ARB therapy, we exclusively used ARB, Losartan, in the preclinical study. This choice was based on our recent research demonstrating Losartan's efficacy in restoring RAS balance and mitigating spike protein-induced cerebrovascular dysfunction (2). We acknowledge that including ACE inhibitor therapy in our preclinical model would offer a more comprehensive reflection of the clinical arm. However, incorporating ACE inhibitors would require a complete repetition of the whole preclinical study, which is unfeasible. The following statement was included in the manuscript to reflect this limitation.

“Future studies should address the limitation of our preclinical model, which exclusively utilized ARB therapy, by incorporating ACEI to mimic the clinical arm where patients received either ARB or ACEI”

Reviewer #2: The paper is well written and addresses an important question of the timing of RAS modulators in COVID-19 related cognitive dysfunction. They show statistically better outcomes in patients receiving the medication post infection. I think the paper is acceptable in its current form. 

Your feedback is greatly appreciated, and we're grateful for your time.

Reviewer #3: In this clinical/preclinical paper, the authors make an attempt to correlate RAS-inhibition before or post onset of Covid-19 with neurological dysfunction. While the preclinical part provides interesting results, appears to be well done and methodologically sound, I have major concerns with respect to the clinical part:

In general, the observed increase in neurological dysfunction in patients on RAS inhibition may be simply due to the fact that these patients were more severely ill, had higher BP levels to start with or cardio-renal problems etc. which afforded additional treatment with RAS inhibitors. 

- The number of patients is relatively small. Meaningful statistics are difficult with such small numbers. - The proportion of black patients is too high to be representative for the US population, let alone the rest of the world. 

- Data on comorbidities are lacking. 

– No information on BP control 

- No information on the methods to evaluate mental status - Coincidence of neurological problems: The number (91) with neurological dysfunction in RAS inhibitor- treated patients does not distinguish between i) one dysfunction in one patient or ii) several neurological problems in one patient. 

We would like to extend our sincere gratitude for the time and effort the reviewer dedicated to reviewing our manuscript. The reviewer’s feedback has been instrumental in strengthening our manuscript. Below is our response to the reviewer’s comments

Firstly, we acknowledge that the clinical part of the study is a retrospective chart review, which inherently relies on electronic medical record documentation of subjective complaints. While we tried to identify objective measures of neurological dysfunction, such as CT scans or MRIs, not all patients underwent such evaluations due to various clinical reasons.

In response to your concern regarding the potential influence of patients' overall clinical condition, including severity of illness, baseline blood pressure levels, or cardiorenal problems, we have implemented several measures to address this issue. Specifically, we have incorporated additional data on comorbidities and blood pressure control into our analysis. These findings indicate that the two groups exhibit similarities in these parameters, thereby indicating that the observed disparities in neurological dysfunction are less likely to be solely attributable to baseline clinical characteristics.

We also acknowledge the high proportion of African American patients included in our study. As residents of the Southeast, where approximately half of the population is African American, it is not surprising that a significant number of African American patients were evaluated in our retrospective chart review. We appreciate your understanding of this demographic representation and assure you that it does not impact the validity of our findings.

Minor problems: 

- line 318: Ang II has the same affinity to AT1R and AT2R. The difference comes rather from the number of receptors in a given tissue under given circumstances.

We apologize for this mistake, and it was corrected in the manuscript to the following “Ang II binds with two main receptors, AT¬1R and AT2R, and triggers different biological responses.”

 - line 320: The AT2R-axis of the "protective RAS" needs to be referenced, e.g. Steckelings et al. Pharm. Reviews 2022. 

The review is being referenced in the manuscript.

- line 323: The protective RAS is not only constituted by the Ang 1-7/Mas axis but also via the AT2R. Stimulation of the unopposed AT2R could be a mechanism of the protective effects of Losartan post SP in the animals. 

The following challenges were made to enhance the clarity of the manuscript:

However, AT2R activation contributes to more vascular protective effects such as vasodilation and anti-inflammatory effects (3). Moreover, ACE-2 is an enzyme that catalyzes the degradation of Ang II to Ang-(1-7), and the resultant molecules activate the Mas receptor which results in protective actions (4-6). Therefore, Ang-(1-7)/Mas axis and the AT2R constitute the protective arm of the RAS.

Additionally, unopposed AT2R stimulation could be a mechanism underlying the protective effect of Losartan demonstrated in the study.

- line 348: Meta-analysis on Covid-19/ RAS inhibition by Gnanenthiran SR et al., J Am Heart Assoc. 2022 should be included.

The Meta-analysis was included in the manuscript.

In 2022, Gnanenthiran et al. published a meta-analysis to assess the safety and efficacy of RAS inhibiton in adults with COVID‐19. The authors found no significant difference in all-cause mortality between patients receiving RAS inhibition = therapy and those without it. However, there was a borderline decrease in acute myocardial infarction among patients on RASi. On the other hand, there was an increased risk of acute kidney injury (AKI) associated with RASi use, particularly in hospitalized patients. Despite this increased risk of AKI, there was no increase in the need for dialysis or other adverse outcomes such as congestive cardiac failure, stroke, or venous thromboembolism (7).

1. Munoz-Fontela C, Dowling WE, Funnell SGP, Gsell PS, Riveros-Balta AX, Albrecht RA, et al. Animal models for COVID-19. Nature. 2020;586(7830):509-15.

2. Burnett FN, Coucha M, Bolduc DR, Hermanns VC, Heath SP, Abdelghani M, et al. SARS-CoV-2 Spike Protein Intensifies Cerebrovascular Complications in Diabetic hACE2 Mice through RAAS and TLR Signaling Activation. Int J Mol Sci. 2023;24(22).

3. Steckelings UM, Widdop RE, Sturrock ED, Lubbe L, Hussain T, Kaschina E, et al. The Angiotensin AT(2) Receptor: From a Binding Site to a Novel Therapeutic Target. Pharmacol Rev. 2022;74(4):1051-135.

4. Raffai G, Durand MJ, Lombard JH. Acute and chronic angiotensin-(1-7) restores vasodilation and reduces oxidative stress in mesenteric arteries of salt-fed rats. Am J Physiol Heart Circ Physiol. 2011;301(4):H1341-52.

5. Santos RA, Campagnole-Santos MJ, Andrade SP. Angiotensin-(1-7): an update. Regul Pept. 2000;91(1-3):45-62.

6. Khajehpour S, Aghazadeh-Habashi A. Targeting the Protective Arm of the Renin-Angiotensin System: Focused on Angiotensin-(1-7). J Pharmacol Exp Ther. 2021;377(1):64-74.

7. Gnanenthiran SR, Borghi C, Burger D, Caramelli B, Charchar F, Chirinos JA, et al. Renin-Angiotensin System Inhibitors in Patients With COVID-19: A Meta-Analysis of Randomized Controlled Trials Led by the International Society of Hypertension. J Am Heart Assoc. 2022;11(17):e026143.

---

## [Decision Letter · Decision Letter 1]

26 Apr 2024

PONE-D-24-03599R1Timing Matters in the Use of Renin-Angiotensin System Modulators and COVID-Related Cognitive and Cerebrovascular DysfunctionPLOS ONE

Dear Dr. Coucha,

Thank you for submitting your manuscript to PLOS ONE. After careful consideration, we feel that it has merit but does not fully meet PLOS ONE’s publication criteria as it currently stands. Therefore, we invite you to submit a revised version of the manuscript in which the clinical part is deleted.

We look forward to receiving your revised manuscript.

Kind regards,

Michael Bader

Academic Editor

PLOS ONE

Reviewers' comments:

Reviewer's Responses to Questions

**Comments to the Author**

1. If the authors have adequately addressed your comments raised in a previous round of review and you feel that this manuscript is now acceptable for publication, you may indicate that here to bypass the “Comments to the Author” section, enter your conflict of interest statement in the “Confidential to Editor” section, and submit your "Accept" recommendation.

Reviewer #3: (No Response)

2. Is the manuscript technically sound, and do the data support the conclusions?

Reviewer #3: Partly

3. Has the statistical analysis been performed appropriately and rigorously? 

Reviewer #3: N/A

4. Have the authors made all data underlying the findings in their manuscript fully available?

Reviewer #3: Yes

5. Is the manuscript presented in an intelligible fashion and written in standard English?

Reviewer #3: Yes

6. Review Comments to the Author

Reviewer #3: Improvements regarding minor concerns acknowledged.

Major concern still with respect to

-small number of patients

- high proportion of black patients with different responses to RAS inhibitors different from white and Asian populations

- rigorous testing not in all patients

7. PLOS authors have the option to publish the peer review history of their article (what does this mean?). If published, this will include your full peer review and any attached files.

Reviewer #3: No

---

## [Author Response · Author response to Decision Letter 1]

2 May 2024

The authors would like to thank all the reviewers (1, 2, and 3) for their invaluable feedback and constructive criticism of our manuscript titled "Timing Matters in the Use of Renin-Angiotensin System Modulators and COVID-Related Cognitive and Cerebrovascular Dysfunction." Their time and expertise are greatly appreciated.

I am writing in response to Reviewer #3's feedback on our study. The authors appreciate Reviewer #3's time and effort in providing a thorough evaluation of our study.

We understand Reviewer #3's concerns regarding the clinical part of our study and the potential limitations associated with it. However, we respectfully disagree with the suggestion to delete the clinical part of the study. We believe that the clinical data significantly contribute to the overall findings and provide valuable insights into the correlation between RAAS inhibition and neurological dysfunction in patients with COVID-19. The clinical data strengthens the scientific merit of our research.

Here is a detailed response to Reviewer #3's concerns regarding the small sample size, the high proportion of African Americans, and the lack of rigorous testing in all patients.

Small sample size: We understand the concerns raised by the reviewer regarding the sample size. However, we would like to emphasize that the goal of this study was to shed light on the potential correlation between RAAS inhibition and neurological dysfunction following COVID-19 exposure. In this study, we tried to identify areas for further investigation. While we acknowledge the importance of having a sufficient number of patients in the clinical study, the findings that we observed based on the 295 patients included in the study can serve as a foundation for future research projects with a larger sample of patients.

Lack of rigorous testing in all patients: Firstly, we acknowledge that the clinical part of the study is a retrospective chart review, which inherently relies on electronic medical record documentation of subjective complaints. We acknowledge the concern raised regarding the absence of comprehensive neurological evaluations, such as CT scans or MRIs, in all patients within our retrospective chart review. It is important to note that the decision for these objective imaging tests was subject to the discretion of attending physicians based on clinical indications and resource availability. Given the retrospective nature of our study and the diversity of patient presentations, not all individuals underwent the same diagnostic procedures. While this may introduce variability in the data collection process, it does not compromise the integrity of our findings. We analyzed the available clinical data to identify trends and associations, ensuring transparency in our reporting of both objective and subjective measures of neurological dysfunction. Furthermore, our study's primary focus was on exploring potential correlations between RAAS inhibition and neurological outcomes in the context of COVID-19, rather than exclusively relying on imaging findings. We have acknowledged this limitation in our revised manuscript and provided 

interpretations of our results within the framework of available clinical data. Despite the variability in diagnostic evaluations, our study still provides valuable insights into the relationship between RAAS modulation and neurological dysfunction post-COVID-19 exposure, warranting further investigation.

High proportion of African Americans: We acknowledge the high proportion of African American patients included in our study. The data was collected from St. Joseph’s/Candler Health System located in Savannah, Georgia. “In 2021, there were 1.41 times more Black or African American (non-Hispanic) residents (76k people) in Savannah, GA, than any other race or ethnicity. There were 54k White (non-Hispanic) and 4.01k White (Hispanic) residents, the second and third most common ethnic groups [1]”. Therefore, it is not surprising that a significant number of African American patients were evaluated in our retrospective chart review. We appreciate your understanding of this demographic representation and assure you that it does not impact the validity of our findings. To provide further clarity, we have included a table detailing the distribution of African American and White patients included in our study. As shown in the table, the majority of the patients were White. While we acknowledge the high proportion of African American patients included in our study; we believe that this demographic diversity enriches our findings and underscores the importance of considering racial disparities in healthcare outcomes.

[1] https://datausa.io/profile/geo/savannah-ga#race_and_ethnicity

Characteristic ACEI/ARB prior to admission (n=177) No ACEI/ARB prior to admission (n=118)

Caucasian 97 (55%) 68 (58%)

African American 78 (44% 44 (37%)

Other 2 (1%) 6 (5%)

We have revised the manuscript to clarify those issues, highlight any limitations within the study, and underscore the need for further studies to validate the findings. We believe that those changes will strengthen the quality and impact of our manuscript.

We respectfully request that the journal reconsider its decision to remove the clinical part of our study.

---

## [Editor Report · Decision Letter 2]

7 May 2024

Timing Matters in the Use of Renin-Angiotensin System Modulators and COVID-Related Cognitive and Cerebrovascular Dysfunction

PONE-D-24-03599R2

Dear Dr. Coucha,

We’re pleased to inform you that your manuscript has been judged scientifically suitable for publication and will be formally accepted for publication once it meets all outstanding technical requirements.

Kind regards,

Michael Bader

Academic Editor

PLOS ONE
---

## [Editor Report · Acceptance letter]

23 Jun 2024

PONE-D-24-03599R2 

PLOS ONE

Dear Dr. Coucha, 

I'm pleased to inform you that your manuscript has been deemed suitable for publication in PLOS ONE. Congratulations! Your manuscript is now being handed over to our production team.

Kind regards, 

on behalf of

Prof. Michael Bader 

Academic Editor

PLOS ONE